# Clinicopathological Features of Non-Small Cell Lung Carcinoma with NRAS Mutation

**DOI:** 10.3390/jpm15050199

**Published:** 2025-05-16

**Authors:** Andrea Ambrosini-Spaltro, Claudia Rengucci, Laura Capelli, Elisa Chiadini, Chiara Bennati, Angelo Delmonte, Silvia Vecchiarelli, Francesco Limarzi, Sofia Nosseir, Graziana Gallo, Mirca Valli, Paola Ulivi, Daniele Calistri

**Affiliations:** 1Pathology Unit, Morgani-Pierantoni Hospital, AUSL Romagna, 47121 Forlì, Italy; francesco.limarzi@auslromagna.it; 2Biosciences Laboratory, IRCCS Istituto Romagnolo per lo Studio dei Tumori (IRST) “Dino Amadori”, 47014 Meldola, Italy; claudia.rengucci@irst.emr.it (C.R.); laura.capelli@irst.emr.it (L.C.); elisa.chiadini@irst.emr.it (E.C.); paola.ulivi@irst.emr.it (P.U.); daniele.calistri@irst.emr.it (D.C.); 3Oncology Unit, Santa Maria Delle Croci Hospital, AUSL Romagna, 48121 Ravenna, Italy; chiara.bennati@auslromagna.it; 4Department of Medical Oncology, IRCCS Istituto Romagnolo per lo Studio dei Tumori (IRST) “Dino Amadori”, 47014 Meldola, Italy; angelo.delmonte@irst.emr.it; 5Oncology Unit, Infermi Hospital, AUSL Romagna, 47923 Rimini, Italy; silvia.vecchiarelli@auslromagna.it; 6Pathology Unit, Santa Maria Delle Croci Hospital, AUSL Romagna, 48121 Ravenna, Italy; sofia.nosseir@auslromagna.it; 7Pathology Unit, Bufalini Hospital, AUSL Romagna, 47521 Cesena, Italy; graziana.gallo@auslromagna.it; 8Pathology Unit, Infermi Hospital, AUSL Romagna, 47923 Rimini, Italy; mirca.valli@auslromagna.it

**Keywords:** lung, carcinoma, NRAS, NSCLC, NGS, molecular, survival, sarcomatoid

## Abstract

**(1) Background**: NRAS mutations affect fewer than 1% of lung adenocarcinomas. The aim of this study was to describe the clinicopathological features of lung carcinomas with NRAS mutations. **(2) Methods**: A series of NRAS-mutated lung carcinomas was collected from a molecular diagnostic unit (from four hospitals). The cases were analyzed with next-generation sequencing. A log-rank test for overall survival (OS) was calculated. **(3) Results**: NRAS mutations were detected in 14/1948 samples (0.72%) of non-small-cell lung carcinomas from 13 patients (8 males, 5 females). NRAS mutations involved codon 61 in the majority (9/13, 69.2%) of cases. The other NRAS mutations affected codon 12 (2/13, 15.4%), codon 13 (1/13, 7.7%), and codon 142 (1/13, 7.7%). In 7/13 cases, co-alterations in additional genes were also present. Pleomorphic/sarcomatoid features were identified in 3/13 (23.1%) cases, in 2/8 (25.0%) histological specimens, and in 2/5 (40.0%) surgical specimens, respectively. Follow-up data were available in 11/13 cases, with 6 patients deceased. By a log-rank test, patients with NRAS mutations in codon 61 had a better outcome (estimated mean of 32.6 ± 7.1 months) compared to those with other NRAS mutations (estimated mean of 8.7 ± 4.4 months), with a significant difference (*p* = 0.048 for OS). **(4) Conclusions**: Lung carcinomas with NRAS mutation may display pleomorphic or sarcomatoid features. Mutations in codon 61 showed a more favorable prognosis than those in other codons.

## 1. Introduction

Lung carcinoma ranks as the second most prevalent tumor and is the primary cause of cancer-related deaths globally, with an estimated 238,340 new cases and 127,070 deaths in 2023 [1]. The most common histotypes are adenocarcinoma (39% in males, 57% in females) and squamous cell carcinoma (25% in males, 12% in females), followed by small-cell carcinoma (11% in males, 9% in females) and large-cell carcinoma (8% in males, 6% in females) [2]. Non-small-cell lung carcinoma (NSCLC) is a term used to collectively describe histological subtypes other than small-cell lung carcinoma. Although traditional morphological classification is extremely important to initially define the clinical approach, the determination of molecular and immunophenotypic profiles is of paramount importance in addressing specific therapeutic decisions [3]. Mutations in EGFR, ALK, ROS1, and other genes define oncogene-addicted lung cancers that may undergo a specific target treatment [4]. Molecular profiling may also be useful in establishing secondary drug resistance [5]. Similarly, immunophenotyping and immunohistochemical analyses are important for detecting tumors that may respond selectively to immune checkpoint inhibitors [6].

The correlation between pathological and clinical features (so-called clinicopathological correlations) has not been extensively analyzed in the literature, and they could be better defined, especially for newly discovered molecular subtypes. Molecular analysis is of paramount importance and is a necessary step in profiling NSCLC and other tumors [7]. However, clinicopathological correlations may help in the diagnosis and treatment of these specific molecular subtypes for three important reasons: (1) Pathologists may be more familiar with recognizing specific molecular subtypes, identifying them from the beginning at the initial diagnostic step. Traditional morphologic criteria may at least address the initial pathological diagnosis towards some specific molecular subtypes. (2) Clinicians may be more aware of specific molecular subtypes, recognize them better from the beginning, and establish a more specific therapy, knowing their natural history. (3) Recognizing these specific molecular subtypes may also help to establish a connection between traditional morphological classification and molecular/immunophenotypic profiling, which may be useful for better defining the mutational landscape of lung carcinoma.

EGFR mutations have been traditionally described in females and non-smokers, with lepidic growth [8,9]. Interestingly, artificial intelligence studies, which have collected thousands of cases, have seemed to confirm these clinicopathological correlations: TP53 adenocarcinomas have been usually associated with solid growth [10], while EGFR-mutated tumors usually displayed acinar and lepidic growth [11]. Specific mutations have been associated with peculiar morphological phenotypes. ALK-rearranged tumors have been usually described in adenocarcinomas with papillary/micropapillary growth, sometimes with signet ring cells [12]; they usually express both TTF1 and p63 by immunohistochemistry [13]. Solid growth with signed ring cells has also been reported in ROS1 rearranged tumors [14]. KRAS-mutated lung carcinoma has recently gained importance since the introduction of specific therapies targeting G12C mutations [15]. KRAS G12C mutations have been frequently observed in lymphocyte-rich tumors, especially in those with TTF1 negativity and PD-L1 positivity [16]. BRAF mutations have been reported in tumors with papillary/micropapillary patterns and sometimes with clear cell changes [17]. Recently, we have described clinicopathological correlations of NSCLCs with BRAF mutations.

Few studies have been conducted on NSCLCs with NRAS mutations [18,19,20]. NRAS mutations are particularly frequent in melanoma, accounting for 15–25% of all melanomas [21,22]. NRAS mutations activate the MAPK pathway; in this, they are similar to BRAF mutations but using CRAF rather than BRAF. NRAS may be blocked with MAPK inhibition by a single agent or by a combination therapies. MAPK inhibitors include selumetinib, trametinib, cobimetinib, and binimetinib [23]. The efficacy of single agents is limited; however, combination therapy usually performs better, especially with BRAF inhibitors. A recent drug, Mb24, has been developed as a selective inhibitor of NRAS [24]. No specific NRAS inhibitor has been approved, but promising results have emerged with naporafenib, an inhibitor of KRAS and NRAS-mutated tumors (in lung carcinoma and melanoma) [25].

Given the potential and important role played by NRAS, the aim of this study was to describe the clinicopathological features of NSCLC patients with NRAS mutations.

## 2. Materials and Methods

We collected all cases of NSCLCs that underwent routine molecular analysis and showed NRAS mutations. All cases were analyzed using next-generation sequencing (NGS) at the Molecular Diagnostics Laboratory of IRST-IRCCS (Meldola, Italy) from 2019 to March 2024. Cases were provided by the 4 Hospital Centers (Cesena, Forlì, Ravenna, and Rimini, Italy) and from external consultations of the same 4 Hospitals. This study was approved by the local Ethical Committee (Comitato Etico della Romagna, CEROM; study ID N. 2771, N. IRST B122 OSCAR, protocol N. 925/2022 I.5/4) on 21 January 2022; the approval was expressed by 23 members of the committee convened during the same session on 21 January 2022. A dedicated pathologist carefully selected suitable material from each case, ensuring that the neoplastic content comprised at least 500 cells and that the ratio of neoplastic to normal tissue was no less than 30%.

### 2.1. Clinical Data

Patient clinical information was obtained from electronic databases. Survival data were also collected from hospital databases. We primarily focused on survival analysis, assuring the specificity of mortality cause. Data on therapeutic regimens were collected if available. However, they did not affect the survival analysis.

### 2.2. Pathological Data

All cases were reviewed by at least two pathologists with special experience in pleuropulmonary pathology. Growth patterns, sarcomatoid features, and all morphological parameters were recorded, especially in the histopathological specimens. Cytological specimens were less informative in terms of growth patterns but were also critically reviewed. The cases were reviewed by at least one pathologist who observed traditional tissue section slides. The second pathologist reviewed the cases remotely with digital slides that were previously scanned using the Aperio AT2 DX System (Leica Biosystems, Nussloch, Germany).

### 2.3. Molecular Analysis

Two main NGS panels were used: the first was from 2019 to June 2022, the second from July 2022 to March 2024. The first NGS panel, conducted from 2019 to June 2022, as previously described [26], was the OncomineTM Focus Assay (Thermo Fisher Scientific, Waltham, MA, USA), an amplicon-based DNA/RNA NGS panel that was able to identify 52 genes. The DNA panel was capable of identifying hotspot mutations across 35 genes (including NRAS) and detecting copy number variants in 19 genes. Meanwhile, the RNA panel was able to identify fusion drivers in 23 genes. The second NGS panel, conducted from July 2022 to March 2024, was the Oncomine™ Comprehensive Assay v3 (OCAv3) (Thermo Fisher Scientific); this amplicon-based DNA/RNA NGS assay targeted 161 cancer-associated genes, including 87 genes with hotspot mutations (like NRAS), 43 genes with focal CNV gains, 48 genes with full CDS for DEL mutations, and 51 gene fusion drivers. The library was prepared using the Oncomine™ Comprehensive Assay v3 DNA/RNA Chef-Ready panel, specifically designed for the Ion Chef™, following the manufacturer’s guidelines. Both NGS panels covered all NRAS hot-spot mutations. Only cases with NRAS mutations were included in the study.

In both panels, we employed the same technique for extracting DNA and RNA, utilizing the MagMAX FFPE DNA/RNA Ultra Kit (Applied Biosystems, Waltham, MA, USA). We adhered to the manufacturer’s guidelines, using either formalin-fixed and paraffin-embedded (FFPE) material (consisting of 6–8 tumor sections, each 5 μm thick) or cytological smears. The DNA concentration was measured using fluorometric quantitation with a Qubit 4.0 Fluorometer and a Qubit DNA HS (High Sensitivity) Assay Kit (Thermo Fisher Scientific).

DNA library preparation was conducted automatically with the “Ion Chef™ System” (Thermo Fisher Scientific), following the manufacturer’s instructions, with 10 ng of input DNA per sample.

The template was then prepared using the Ion Chef™ System, integrating DNA and RNA from the same sample on a single chip, and sequencing was performed on the Ion S5 Plus platform (Thermo Fisher Scientific) using Ion 520 or 540 Chips (Thermo Fisher Scientific).

The initial quality control assessment was performed using Torrent Suite Software™ (version 5.12.3), which evaluated chip loading density, the average length of reads, and number of mapped reads. Each sample was subsequently analyzed with Ion Reporter™ Software (version 5.16), a specific bioinformatic tool for variants, filtering, and annotations.

Variants were reported only if they exhibited a variant allele frequency (VAF) of 5% or higher and had a coverage exceeding 500X. Furthermore, variants with significant therapeutic relevance, based on the sample pathologies in which they have been detected, were also described.

### 2.4. Immunohistochemistry

We utilized an automated immunostainer (ULTRA, Ventana Medical Systems, Roche, Tucson, AZ, USA) with the following antibodies: VENTANA anti-ALK (D5F3) rabbit monoclonal primary antibody (Ventana Medical Systems, AZ, USA), ROS1 (D4D6) rabbit monoclonal antibody (Cell Signaling Technology, Inc., Danvers, MA, USA), and PD-L1 Dako 22C3 anti-PD-L1 primary antibody (Agilent, Santa Clara, CA, USA). The PD-L1 staining was a laboratory-based assay that yielded results consistent with those documented in the literature, as previously noted [27].

For immunohistochemical analyses, we used 4 μm thick sections, which were placed on positively charged slides. We used EZ Prep solution (Ventana Medical Systems) to remove paraffin and a reaction buffer to rinse the slides among the staining steps. Antigen retrieval was conducted using Cell Conditioning 1 (CC1) (pH 8.0) antigen retrieval solution (Ventana Medical Systems) for 64 min at 95 °C. For ALK, specimens were incubated with a prediluted primary anti-ALK antibody for 16 min using OptiView DAB Detection and Amplification system. For ROS1, specimens were incubated with primary anti-ROS1 antibody at a concentration of 1:100 for 32 min. For PD-L1, specimens were incubated with primary anti-PD-L1 antibody at a concentration of 1:25 for 64 min at 37 °C, followed by the application of an OptiView DAB IHC Detection Kit. The slides underwent hematoxylin staining and were covered with coverslips. Each run included appropriate positive controls. For PD-L1 staining, a normal tonsil served as an internal control on each slide.

Staining was assessed using a specific scoring system, based on the antibody applied. ALK cases were classified as positive or negative, following the manufacturer’s guidelines [28]. ROS1 cases were evaluated for moderate (2+) or strong (3+) cytoplasmic staining intensity, with at least 50% neoplastic cells [29]. PD-L1 cases were deemed adequate if there were at least 100 tumor cells present. A qualified pathologist reviewed the slides, and challenging cases were discussed collegially. According to the tumor proportion score (TPS), PD-L1 cases were categorized into three groups: (1) less than 1% positive cells (TPS < 1%, negative); (2) 1–49% positive cells (TPS: 1–49%, low expression); and (3) 50% or more positive cells (TPS ≥ 50%, high expression) [30].

### 2.5. Statistical Analysis

Kaplan–Meier survival curves were utilized to estimate overall survival (OS). OS was defined as the duration from the date of molecular analysis—which aligns with the diagnosis of metastatic or advanced disease—to either the date of death or the last follow-up for patients who were still alive. We used the log-rank test to compare OS between the different subgroups. The median OS, along with a 95% confidence interval (CI), was determined.

We investigated the differences between NRAS mutations at codon 61 and those at other codons, primarily 12, 13, and 142. The Mann–Whitney test for non-parametric variables was calculated to compare variables across different categories. A *p*-value of 0.05 was set as the threshold for statistical significance, using a 2-tailed hypothesis. Statistical analyses were conducted using Microsoft Excel 2020 (Microsoft Corp., Redmond, WA, USA) and SPSS, version 25 (IBM Corp., Armonk, NY, USA).

## 3. Results

In the period considered (beginning 2019 to March 2024), 14 NSCLC samples with NRAS mutations from 13 patients were analyzed. During the same period, 1948 NSCLC samples were tested. Thus, the NRAS samples represented 14/1948 and 0.72% of all the NSCLCs analyzed. For practical and statistical purposes, repetitive analyses with positive NRAS mutations in the same patient were excluded. One patient had two positive NRAS samples (one cytological and one histological), from which one sample (the cytological one) was excluded. Finally, 13 cases from 13 patients were included in the study.

### 3.1. Clinical Features

Patients affected by NSCLCs with NRAS mutations were more frequently male (8/13, 61.5%) than female (5/13, 38.5%). Age varied from 46 to 83 years, with a mean of 67.2 ± 11.3. Additional clinical data are presented in Appendix A.

Follow-up was available for 11/13 cases, varying from 1 to 48 months (median: 14 months). Six patients were deceased, while seven patients were alive.

### 3.2. Pathological and Molecular Features

Samples analyzed for molecular analysis included cytological smears in 4/13 (30.8%), cytological cell blocks in 1/13 (7.7%), histological biopsy in 4/13 (30.8%), and histological surgical specimens in 4/13 (30.8%). Collectively, most cases were histological specimens (8/13, 61.5%), whereas only a few were cytological (5/13, 38.5%).

The cases were mainly adenocarcinomas (8/13, 61.5%), followed by NSCLC with sarcomatoid features (3/13, 23.1%), NSCLC not otherwise specified (NOS) (1/13, 7.7%), and squamous cell carcinoma (1/13, 7.7%). Sarcomatoid features were particularly interesting and were detected in 3/13 (23.1%) cases, in 2/8 (25.0%) considering only histological specimens, and in 2/5 (40.0%) considering only surgical specimens where more tissue was available for morphological analysis. Sarcomatoid features are shown in Figure 1.

NRAS mutations involved codon 61 in the majority (9/13, 69.2%) of cases. The other NRAS mutations were present in codon 12 (2/13, 15.4%), codon 13 (1/13, 7.7%) and codon 142 (1/13, 7.7%). Collectively, NRAS mutations were grouped into codon 61 (9/13, 69.2%) and non-codon 61 (4/13, 30.8%). Figure 2 illustrates a pie chart of the distribution of NRAS mutations.

The main molecular features of NRAS-positive cases are summarized in Table 1.

Co-alterations were detected in 7/13 cases and predominantly involved the following genes: STK11 in three cases, MET in two cases, TP53 in one case, and RET in one case. Sarcomatoid features, as previously described, were detected in three cases (cases N. 2, 10, 11), among which two displayed co-alterations (cases N. 2 and 11). Detailed information on the detected NRAS co-alterations is presented in Table 2.

Immunohistochemical analyses were available in 10/13 cases for PD-L1 and in 8/13 cases for ALK and ROS1. PD-L1 was negative in two cases, with low expression in seven cases and high expression in one case. ALK and ROS were both negative in all eight cases examined.

### 3.3. Statistical Analyses

Survival analysis revealed that patients with NRAS mutations in codon 61 performed better than those with other NRAS (non-codon 61) mutations. Specifically, cases with codon 61 mutations showed an estimated mean of 32.6 ± 7.1 months, with a 95% confidence interval range of 18.6–46.5 months, whereas cases with non-codon 61 mutations showed an estimated mean of 8.7 ± 4.4 months, with a 95% confidence interval range of 0.1–17.2 months. The log-rank test showed that this difference was statistically significant (chi-square: 3.927; degrees of freedom: 1; *p* = 0.048 for OS). Further information from the survival analysis are summarized in Appendix A.

Survival curves are shown in Figure 3.

No significant differences were found between codon 61 and non-codon 61 NRAS mutations in terms of age (*p* = 0.49), sex (*p* = 0.58), type of material examined (*p* = 0.52). and histological subtype (*p* = 0.54). This indicates that cases with codon 61 mutations and those with non-codon 61 mutations did not display important clinicopathological differences other than their type of mutation at initial presentation.

## 4. Discussion

This study demonstrates the following:NRAS mutations were frequently associated with sarcomatoid features.NRAS mutations in codon 61 were more frequent and displayed better prognosis than other NRAS mutations, being this difference statistically significant in OS analysis.

RAS proteins are members of the small GTPase protein family and have three members: KRAS, HRAS, and NRAS [31]. KRAS is the most frequently mutated, while NRAS and HRAS are seldom altered [32]. KRAS is one of the most frequently mutated genes in solid tumors and is mutated not only in the lung but also in the colon and the pancreas [33]. KRAS plays a central role in signal transduction and has gained considerable attention in recent years as it has become a target of specific drugs, rendering it extremely important in different therapeutic strategies [34]. New drugs (sotorasib and adagrasib) are specifically effective in tumors harboring the KRAS G12C mutation [22]. NRAS mutations are typically found in melanomas, the colon, the thyroid, and bone marrow [35]. Melanomas harboring NRAS mutations represent 15–25% of all melanomas [21,22] and behave more aggressively than other melanomas [36]. Mutated NRAS triggers the MAPK signaling cascade through the activation of RAF [37]. Several drugs have been developed to treat NRAS mutations in melanomas [36]. In the lung, naporafenib is a specific NRAS inhibitor that has shown promising results [25].

In the landscape of driver mutations in NSCLC, NRAS mutations are rare and account for only 1% of all driver mutations. Three studies examined the features of NRAS-mutated lung adenocarcinoma. Sasaki [20] found only 1 case with NRAS mutation over 195 lung adenocarcinomas, underlining the rarity of these mutations. Ohashi [19] collected 30 cases with NRAS mutations among 4562 patients. However, both studies [19,20] were conducted several years before the routine introduction of the NGS platform. NGS platforms have revolutionized molecular analyses and clinical practice as they are capable of analyzing different biomarkers simultaneously [38]. Only a recent article by Dehem et al. analyzed the molecular and pathological features of NRAS-mutated lung carcinomas using NGS [18].

Our study was conducted using NGS platforms and is the first to show that NRAS-mutated lung carcinomas may exhibit sarcomatoid features. This may be of pathological and clinical importance because pathologists and oncologists facing sarcomatoid carcinoma may be aware that sarcomatoid features could be associated with NRAS mutations, among other well-known mutations. Sarcomatoid carcinomas of the lung are usually associated with mutations in TP53, KRAS, EGFR, and MET [39]. MET alterations have been extensively analyzed in the literature because exon 14 (METex14) skipping mutations can be targeted by specific drugs [40]. In our study, sarcomatoid features were found in 3/13 (23.1%) cases, which is a limited number of cases. However, this percentage was higher when considering tissues with more available pathological material. Sarcomatoid features were found in 2/8 (25.0%) cases considering only histological specimens, and in 2/5 (40.0%) cases considering only surgical specimens, in which much more tissue was available for morphological analysis.

In 2/3 (66.7%) cases, sarcomatoid features were also associated with co-mutations; however, in none of them, well-known alterations of sarcomatoid carcinoma were detected (c-met or p53 mutations). Sarcomatoid carcinoma in the lung may be associated with two important molecular groups: one cluster is triggered by tobacco and typically involves RAS mutations, the MAPK pathway, and PD-L1 positivity; the other cluster is usually associated with targetable molecular alterations such as c-met [41]. A substantial group of sarcomatoid carcinomas has been reported to belong to the first cluster and carries KRAS mutation and high PD-L1 expression [42]. These data underline the important role played by RAS mutations (including NRAS mutations) in eventually inducing the development of sarcomatoid features.

This study also analyzed the prognostic impact of a specific NRAS codon mutation on OS. All previous studies agreed that codon 61 mutations are the most frequent. Ohashi [19] found codon 61 mutations in 80% of cases and Dehem [18] in 69.5%. However, in prognostic analysis, Dehem et al. did not find a significant difference in OS according to the type of mutation (codon 61 vs. other) [18], whereas our study found a significant difference, with mutated cases involving codon 61 showing a better prognosis. This feature may also be useful in clinical practice because mutations in codons other than codon 61 may be treated more aggressively. However, these prognostic data should be examined in future studies involving more patients.

The limitations of this study are mainly related to the small number of cases, particularly for survival analysis. However, the described features were statistically significant and may be clinically important. Further studies with more cases are necessary to obtain more robust results. Another limitation is the absence of progression-free survival (PFS) analysis. However, in this series, different therapeutic strategies and clinical presentations made subtle alterations in PFS extremely variable. Therefore, we decided to focus only on overall survival and its role in determining the final prognostic impact. Another possible limitation is the absence of stage in the survival analysis. Stage is undoubtedly an important prognostic factor. However, in our institution, patients are usually tested for molecular analysis only in advanced/metastatic stages (either for the beginning or after a period free of disease); therefore, we do not think that in this particular context, stage would be so informative, and its absence might not be substantial.

## 5. Conclusions

This study shows that NRAS-mutated lung carcinoma may exhibit sarcomatoid features, rendering sarcomatoid features particularly important for detection. NRAS mutations should be clearly described in the molecular pathology report. Furthermore, NRAS-mutated cases with NRAS mutations specifically involving codon 61 show better prognosis than those with other NRAS mutations, making the distinction between codon 61 and non-codon 61 useful in clinical practice.

## Figures and Tables

**Figure 1 jpm-15-00199-f001:**
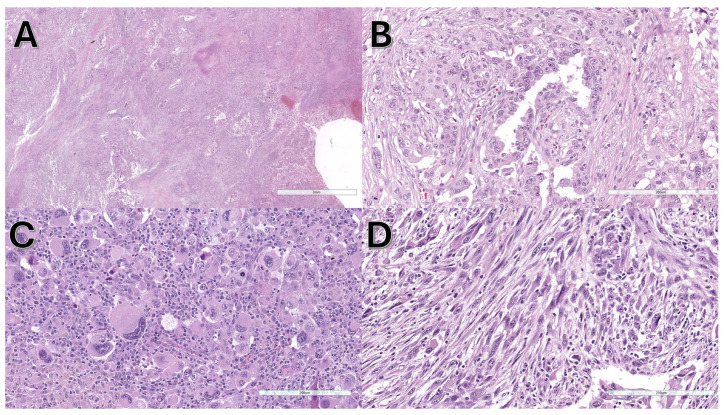
Morphological features of a lung adenocarcinoma with NRAS mutation and sarcomatoid features. (**A**) At low power, the neoplasia shows diffuse growth and necrotic areas. (**B**) In some areas, the atypical glands are evident, and the diagnosis of adenocarcinoma is straightforward. In other areas, highly pleomorphic giant cells (**C**) and highly atypical spindle cells (**D**) are observed, and the diagnosis of pleomorphic carcinoma with sarcomatoid features is necessary.

**Figure 2 jpm-15-00199-f002:**
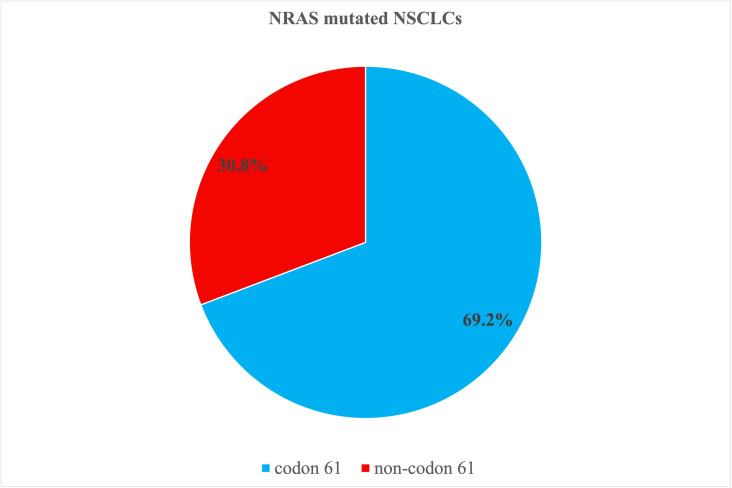
The pie graph illustrates the NRAS mutations in the series. Mutated cases involving codon 61 represent the majority (69.2%) of cases.

**Figure 3 jpm-15-00199-f003:**
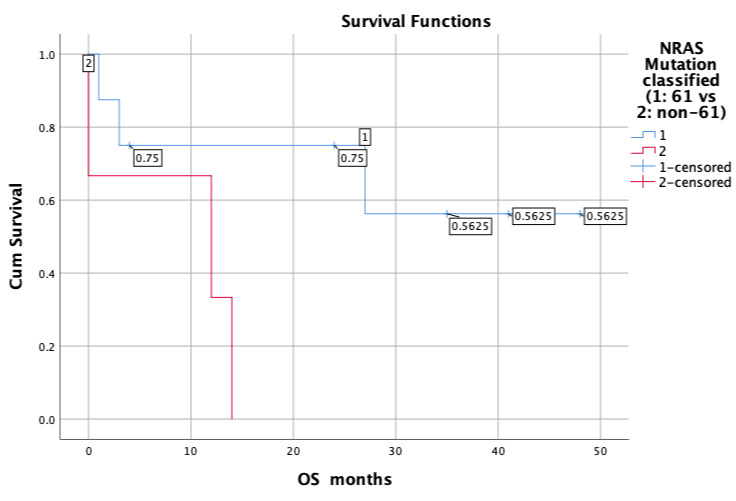
Overall survival analysis illustrates that cases with NRAS mutations in codon 61 (blue line, group 1) show better prognosis than those with other NRAS (non-codon 61) mutations (red line, group 2). The log-rank test revealed a significant difference between the two groups (*p* = 0.048).

**Table 1 jpm-15-00199-t001:** Summary of all identified NRAS mutations.

Codon		N. (%)	Mutation	Exon	N.
Codon 61		9 (69.2%)			
			c.182A > T, p.(Gln61Leu)	3	4
			c.181C > A, p.(Gln61Lys)	3	3
			p.Q61L (c.182 A > T)	3	1
			c.182A > G, p.(Gln61Arg), (Q61R)	3	1
Non-codon 61		4 (30.8%)			
	codon 12	2 (15.4%)	c.35G > C, p.(Gly12Ala)	2	1
			c.425T > C, p.(Ile142Thr)	4	1
	codon 13	1 (7.7%)	c.38G > A, p.(Gly13Asp)	3	1
	codon 142	1 (7.7%)	c.425T > C, p.(Ile142Thr)	4	1
Total		13			

**Table 2 jpm-15-00199-t002:** Detailed information on the detected co-alterations, with data on sarcomatoid features and PD-L1. OS status: 0 alive, 1 deceased. Legend: OS, overall survival; NA, not available.

Case N.	Sex	Age	Exon	Codon	Co-alterations	Sarcomatoid Features	PD-L1	OS Status	OS Months
1	M	63	3	61	None		NA	1	27
2	M	58	3	13	STK11, CDKN2A, NF1, FGFR2	Y	NA	1	14
3	F	72	3	61	None		Low	0	48
4	M	63	2	12	None		Low	1	12
5	F	66	3	61	None		Neg	0	24
6	F	83	2	12	MET, JAK2		Low	1	0
7	M	46	3	61	RET		High	1	1
8	F	81	4	142	TP53, MET, ATR		Low	NA	
9	F	76	3	61	None		Low	0	35
10	M	51	3	61	None	Y	NA	0	41
11	M	79	3	61	KIT, ATR, RAC1, AR	Y	Neg	0	4
12	M	65	3	61	STK11, EGFR		Low	1	3
13	M	71	3	61	STK11		Low	NA	

## Data Availability

The data that support the findings of this study are available from the corresponding author upon reasonable request.

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
