# Peer review of "Clinicopathological Features of Non-Small Cell Lung Carcinoma with NRAS Mutation"

_jpm, 2025, doi:10.3390/jpm15050199_

Round 1

Reviewer 1 Report

Comments and Suggestions for Authors

Main question addressed by this research is the incidence, clinical, pathological and survival features of NRAS mutations in NSCLC. Topic is relevant, original and much needed as these mutations are extremely rare and literature is scant.

It adds to subject area as this is the first NGS analysis of NRAS mutations.

Conclusions are consistent with evidence and material presented. Tables and figures add depth and references are appropriate.

Molecular analysis is explained in great detail. Survival analysis is very informative and detailed.

Possible improvements:

Males have a higher incidence of squamous cell cancers. NRAS was found more common in males, however these were majority adenocarcinomas. May be informative to stratify smoking status. It appears NRAS mutations happen more commonly in non smokers from the information.

If NRAS mutations have more sarcomatoid features, does it predict a good response to immunotherapy? This is historically the case with lung/ mesotheliomas. How were these patients treated? If we say NRAS patients live longer, it is important to share their PD-L1 status as well as they may have been treated with immunotherapy and this would be a confounder.

Author Response

Thank you very much for your helpful suggestions. We made the following modifications.

Males have a higher incidence of squamous cell cancers. NRAS was found more common in males, however these were majority adenocarcinomas. May be informative to stratify smoking status. It appears NRAS mutations happen more commonly in non smokers from the information.

NRAS mutated lung carcinoma are usually adenocarcinomas, as described by Dehem https://doi.org/10.1016/j.lungcan.2023.107393. Information on smoking status, when available, was added in Supplementary Table 1. Many patients were smokers, or ex-smokers.

If NRAS mutations have more sarcomatoid features, does it predict a good response to immunotherapy? This is historically the case with lung/ mesotheliomas. How were these patients treated? If we say NRAS patients live longer, it is important to share their PD-L1 status as well as they may have been treated with immunotherapy and this would be a confounder.

Information on therapy, when available, was added in Supplementary Table 1. PD-L1 status was added in both Table 2 and Supplementary Table 1.

Reviewer 2 Report

Comments and Suggestions for Authors

The authors aimed to analyze the clinicopathological features of lung carcinomas with NRAS mutations and showed 2 important speculations; 1. NRAS mutations are frequently associated with sarcomatoid features, 2. NRAS mutations in codon 61 are more frequent and associated with better prognosis in OS.

The data and discussion are clear and I felt it is well-written manuscript.

I would like to comment to a few points:

  1. The authors declared that ethical review and approval of their institute and informed consent from the patients were waived because only anonymized clinical data were used in this study. But I do not think it is appropriate. Retrospective studies also should be review and approved by institutional review board. Written informed consent is not always easy to get in such study, but opt-out manner can be performed. Is it a local rule of our institution or nation? Is it all right in the authors’ institution / country?

  1. The authors mentioned only the small number of cases and the absence of PFS analysis as their limitations. Related to that, I think that lack of the data about the stage and treatment of the cancer is also a large limitation. Although they show the difference of OS between codon 61 mutation cases and the others, I cannot evaluate the cofounding of the difference of stage and the difference of treatment strategy due to the stage. If the cases with codon 61 mutation are early cases, this phenomenon itself can be a result of low malignancy of the tumor, so I still think the author’s speculation is appropriate. I only think that they should show the data and discuss about it.

Author Response

Thank you very much for your helpful suggestions. We made the following modifications.

The authors aimed to analyze the clinicopathological features of lung carcinomas with NRAS mutations and showed 2 important speculations; 1. NRAS mutations are frequently associated with sarcomatoid features, 2. NRAS mutations in codon 61 are more frequent and associated with better prognosis in OS. The data and discussion are clear and I felt it is well-written manuscript. I would like to comment to a few points: 1. The authors declared that ethical review and approval of their institute and informed consent from the patients were waived because only anonymized clinical data were used in this study. But I do not think it is appropriate. Retrospective studies also should be review and approved by institutional review board. Written informed consent is not always easy to get in such study, but opt-out manner can be performed. Is it a local rule of our institution or nation? Is it all right in the authors’ institution / country?

Yes, you are right. We sincerely apologize for the inconvenience. Authorization by the ethical committee has however been previously obtained in a larger study which included NSCLCs and also NRAS mutated cases. This has been corrected in the specific paragraph at the end of the study. Written consent was not obtained, but data were completely anonymized, and no personal information was obtained by any of the authors.

2. The authors mentioned only the small number of cases and the absence of PFS analysis as their limitations. Related to that, I think that lack of the data about the stage and treatment of the cancer is also a large limitation. Although they show the difference of OS between codon 61 mutation cases and the others, I cannot evaluate the cofounding of the difference of stage and the difference of treatment strategy due to the stage. If the cases with codon 61 mutation are early cases, this phenomenon itself can be a result of low malignancy of the tumor, so I still think the author’s speculation is appropriate. I only think that they should show the data and discuss about it.

Information on therapy, when available, was added in Supplementary Table 1. Stage was not considered because cases were all in advanced/metastatic stage. Molecular analysis in our Institution is usually performed when patients are in advanced stages or when they progress after previous therapy from lower stages. This limitation however has been added at the end of the Discussion Section (lines 605-610).

Reviewer 3 Report

Comments and Suggestions for Authors

The manuscript titled "Clinicopathological features of non-small cell lung carcinoma with NRAS mutation" investigates the clinicopathological and molecular characteristics of non-small cell lung carcinomas (NSCLCs) harboring NRAS mutations, which are rare and occur in approximately 0.72% of NSCLC cases. Fourteen samples from 13 patients were analyzed using next-generation sequencing (NGS). Most NRAS mutations were located at codon 61, and patients with these mutations demonstrated better overall survival (OS) compared to those with mutations at other codons. Sarcomatoid features were observed in a subset of cases and may be associated with NRAS mutations. The study suggests that codon 61 NRAS mutations are prognostically favorable and may have diagnostic and therapeutic implications. However, several sections of the manuscript would benefit from revision to improve clarity, enhance methodological transparency, and strengthen the overall scientific presentation—thereby increasing the manuscript’s impact and accessibility to a broader scientific audience.

  1. Abstract: Consider briefly noting the survival times for codon 61 vs non-codon 61 mutations for better impact.
  2. Introduction:

2.1 Could you cite more recent studies or reviews supporting the claim that NRAS-targeted therapies are emerging?

2.2  Could the introduction better integrate the clinical-pathological relationship with molecular profiling earlier on?

  1. Method: Would including power calculation or confidence intervals for survival comparisons strengthen the analysis?
  2. Results:

4.1 Does the small number of patients limit the power of the statistical conclusions, particularly for survival analysis?

4.2 Figure 2 and 3 are helpful—would a combined heatmap of mutation + sarcomatoid features + IHC improve data visualization?

5.Discussion: Could the role of NRAS in driving sarcomatoid morphology be speculatively discussed based on MAPK pathway activation?

  1. Conclusion: Could the authors add a brief suggestion for how this data could inform routine NGS panel designs or reporting in lung cancer
  2. Consider merging Table 2 with clinical outcomes for richer insights.
  3. Figure 1: Could higher-resolution or annotated images improve clarity?
  4. Figure 3: Kaplan-Meier OS, Include number at risk per time point below the x-axis.

Author Response

Thank you very much for your helpful suggestions. We made the following modifications.

The manuscript titled "Clinicopathological features of non-small cell lung carcinoma with NRAS mutation" investigates the clinicopathological and molecular characteristics of non-small cell lung carcinomas (NSCLCs) harboring NRAS mutations, which are rare and occur in approximately 0.72% of NSCLC cases. Fourteen samples from 13 patients were analyzed using next-generation sequencing (NGS). Most NRAS mutations were located at codon 61, and patients with these mutations demonstrated better overall survival (OS) compared to those with mutations at other codons. Sarcomatoid features were observed in a subset of cases and may be associated with NRAS mutations. The study suggests that codon 61 NRAS mutations are prognostically favorable and may have diagnostic and therapeutic implications. However, several sections of the manuscript would benefit from revision to improve clarity, enhance methodological transparency, and strengthen the overall scientific presentation—thereby increasing the manuscript’s impact and accessibility to a broader scientific audience.

  1. Abstract: Consider briefly noting the survival times for codon 61 vs non-codon 61 mutations for better impact.

In the abstract estimated mean was added for both groups (codon 61 vs non-codon 61) (lines 30-31).

  1. Introduction:

2.1 Could you cite more recent studies or reviews supporting the claim that NRAS-targeted therapies are emerging?

Specific studies regarding NRAS inhibition have been added (lines 87-91).

2.2 Could the introduction better integrate the clinical-pathological relationship with molecular profiling earlier on?

A more extensive explanation of clinico-patological correlations for lung adenocarcinomas has been added in lines 68-81.

  1. Method: Would including power calculation or confidence intervals for survival comparisons strengthen the analysis?

Yes, we think that confidence intervals could better define statistical analyses. Lines 283-284: “The median OS, along with a 95% confidence interval (CI), was determined.”

  1. Results:

4.1 Does the small number of patients limit the power of the statistical conclusions, particularly for survival analysis?

Yes, the limited number of cases may partly limit the clinical impact. However, the statistical significance was reached. This has been explained in the limitations of the study at the end of Discussion Section (lines 606-608).

4.2 Figure 2 and 3 are helpful—would a combined heatmap of mutation + sarcomatoid features + IHC improve data visualization?

Thank you for the suggestion. Unfortunately, we are not able to merge Figure 2 and 3 and to realize such a heat map at the moment. We do not have the specific software to perform this analysis.

  1. Discussion: Could the role of NRAS in driving sarcomatoid morphology be speculatively discussed based on MAPK pathway activation?

We searched in the literature and found some interesting papers on this topic; we added a short paragraph in the Discussion section (lines 582-588).

  1. Conclusion: Could the authors add a brief suggestion for how this data could inform routine NGS panel designs or reporting in lung cancer

We add the following statement: “NRAS mutation are important to be identified and clearly signaled reported in the molecular pathology report” (lines 623-624).

  1. Consider merging Table 2 with clinical outcomes for richer insights.

In Table 2 information on OS status and OS months have been added.

  1. Figure 1: Could higher-resolution or annotated images improve clarity?

Figure 1 was changed with more specific details and higher quality resolution.

  1. Figure 3: Kaplan-Meier OS, Include number at risk per time point below the x-axis.

Unfortunately, SPSS software does not allow me to insert these number below the x-axis. However, I have inserted these numbers below the graph lines (possible in SPSS). Supplementary Tables S2 and S3 with data on survival analyses were added.

As requested, additional modifications were made in all Sections (Introduction, Materials and Methods, Results, Dicussion and Conclusion) with track changes feature.

Round 2

Reviewer 3 Report

Comments and Suggestions for Authors

Thank you for the opportunity to review this manuscript once again. The authors have revised the manuscript according to all of the reviewers' comments. However, the similarity index remains high at 35%. We would like to kindly request the authors to further reduce the similarity to below 20%.

Author Response

Thank you for the opportunity to review this manuscript once again. The authors have revised the manuscript according to all of the reviewers' comments. However, the similarity index remains high at 35%. We would like to kindly request the authors to further reduce the similarity to below 20%.

Thank you again. Further modifications in the text were performed. Please remind the fact the Materiale and Methods were similar in our previous publications.